# Neutrophils: Underestimated Players in the Pathogenesis of Multiple Sclerosis (MS)

**DOI:** 10.3390/ijms21124558

**Published:** 2020-06-26

**Authors:** Mirre De Bondt, Niels Hellings, Ghislain Opdenakker, Sofie Struyf

**Affiliations:** 1Laboratory of Molecular Immunology, Department of Microbiology, Immunology and Transplantation, Rega Institute for Medical Research, KU Leuven, Herestraat 49—Box 1042, 3000 Leuven, Belgium; mirre.debondt@kuleuven.be; 2Neuro Immune Connections & Repair Lab, Department of Immunology and Infection, Biomedical Research Institute, Hasselt University, Martelarenlaan 42, 3500 Hasselt, Belgium; niels.hellings@uhasselt.be; 3Laboratory of Immunobiology, Department of Microbiology, Immunology and Transplantation, Rega Institute for Medical Research, KU Leuven, Herestraat 49—Box 1044, 3000 Leuven, Belgium; ghislain.opdenakker@kuleuven.be

**Keywords:** neutrophils, multiple sclerosis, autoimmunity, antigen presentation

## Abstract

Neutrophils are the most abundant circulating and first-responding innate myeloid cells and have so far been underestimated in the context of multiple sclerosis (MS). MS is the most frequent, immune-mediated, inflammatory disease of the central nervous system. MS is treatable but not curable and its cause(s) and pathogenesis remain elusive. The involvement of neutrophils in MS pathogenesis has been suggested by the use of preclinical animal disease models, as well as on the basis of patient sample analysis. In this review, we provide an overview of the possible mechanisms and functions by which neutrophils may contribute to the development and pathology of MS. Neutrophils display a broad variety of effector functions enabling disease pathogenesis, including (1) the release of inflammatory mediators and enzymes, such as interleukin-1β, myeloperoxidase and various proteinases, (2) destruction and phagocytosis of myelin (as debris), (3) release of neutrophil extracellular traps, (4) production of reactive oxygen species, (5) breakdown of the blood–brain barrier and (6) generation and presentation of autoantigens. An important question relates to the issue of whether neutrophils exhibit a predominantly proinflammatory function or are also implicated in the resolution of chronic inflammatory responses in MS.

## 1. Neutrophils

Neutrophilic granulocytes are the most abundant cell population (70%) in the circulating leukocyte fraction in the human species. They belong to the polymorphonuclear cell family together with basophils and eosinophils because of their multi-lobular nucleus [1,2]. After their formation in the bone marrow from a common myeloid progenitor cell, they circulate in the bloodstream for 7–10 h, after which they migrate into tissues and exert their functions. These innate immune cells are often seen as ‘first responder cells’ during acute inflammation. They are highly motile cells and were assumed to display a short lifespan of about one day. The latter survival time is currently disputed because neutrophils could still be detected after 7 days under certain conditions [3,4]. Such refinement of detection may be limited to specific subpopulations of neutrophils and yields discussions about future immunophenotyping of neutrophil subsets (*vide infra*). Under physiological conditions, the majority of neutrophils are patrolling cells and just circulate through the body in resting state to avoid the release of their toxic, intracellular content. Activation of neutrophils requires two steps: (1) priming of neutrophils by bacterial products or cytokines/chemokines and (2) binding of activating signals to membrane receptors at the site of injury [5].

In response to an inflammatory stimulus, neutrophils are quickly activated through G-protein-coupled receptors (GPCRs) and migrate along four types of chemical gradients to the site of infection/injury/inflammation, which is called ‘chemotaxis’. These four chemotactic cues may be bacterial formyl peptides, complement molecules C3a and C5a, leukotrienes or chemotactic cytokines, dubbed chemokines [6]. The prototype of human neutrophil chemokine is granulocyte chemotactic protein-1, named interleukin-8, and later still renamed CXC chemokine ligand (CXCL)-8 [7]. CXCL8 uses two CXC chemokine receptors (CXCR): CXCR1 and CXCR2 (*vide infra*). Upon arrival at the target sites, neutrophils are activated and execute their effector functions, mainly intended for the elimination of pathogens. By the release of cytokines and chemokines, neutrophils amplify the inflammatory cascade and attract other immune cells. Neutrophils can act as phagocytes and internalize microorganisms or particles, foremost when these are coated with opsonins, such as antibodies or complement molecules. To kill the engulfed pathogens, reactive oxygen species (ROS) are produced and released into the phagosome. Neutrophils also secrete granules which are classified in one system (on the basis of biosynthesis during the maturation of neutrophils in the bone marrow) in the following three categories: primary or azurophilic granules (containing, for instance, myeloperoxidase (MPO) and the serine proteinases neutrophil elastase (NE), cathepsin G and proteinase 3), secondary or specific granules (containing, e.g., lysozyme, lactoferrin and neutrophil gelatinase-associated lipocalin) and tertiary or gelatinase B granules. In addition to granules, neutrophils also contain smaller secretory vesicles with alkaline phosphatase, cytokines/chemokines and CD35, the latter of which is first released upon activation [8]. The release of this antimicrobial content is called degranulation, in which the order is reciprocal with the synthesis in the bone marrow, namely secretory vesicles and tertiary granules released first and primary granules last. A recently discovered function of neutrophils is the formation of neutrophil extracellular traps (NETs), which includes the release of fiber-like structures of DNA [9]. These NETs trap and eliminate microbes independently of phagocytosis and may also serve as a physical barrier to prevent spreading of pathogens. Neutrophils also act as immunomodulators by promoting the maturation of professional antigen presenting cells (APCs) [10]. Coculturing of neutrophils and dendritic cells (DCs) leads to activation of DCs. Activation is dependent on cell-to-cell contact and is accompanied by an upregulated expression of CD40, CD86 and human leukocyte antigen-DR isotype (HLA-DR) [11]. Another recent proposal is that neutrophils themselves act as alternative APCs and in this way participate in adaptive immunity [12,13]. It is believed that neutrophils influence adaptive immunity through the transport of antigens to lymph nodes (LNs), presentation of antigens to T cells and interaction with APCs. This function implies that such neutrophils may leave the site of antigen capture, enter the lymph circulation and transport and present antigens, together being processes in need of cumulated time intervals which presumably supersede the supposed life-span of only one day [14]. Bone marrow-derived murine neutrophils are able to transform into a DC-like population after treatment with granulocyte-macrophage colony-stimulating factor (GM-CSF), called ‘neutrophil-DC hybrids’ (Figure 1). These DC-neutrophil hybrid cells express markers of both neutrophils (Ly6G, CXCR2) and DCs (major histocompatibility complex class II (MHC II), CD80/86) and develop some APC-like properties. They thus acquire DC characteristics while maintaining intrinsic neutrophil features such as phagocytosis and NET release [15,16]. Regarding the origin of the hybrid population, Geng et al. showed that they originate from immature band cells that either progress into mature polymorphonuclear cells or transdifferentiate into neutrophil–DC hybrids. These authors also demonstrate that the hybrid population is able to present bacterial antigens to CD4^+^ T cells and assist in rapid bacterial clearance in vivo [15,16]. Also, in mouse models for inflammatory diseases, MHC II and CD86 expressing neutrophils, that are able to present antigens to naive CD4^+^ T cells, were isolated [17]. These neutrophil-DC hybrids originate from transdifferentiated neutrophils after extravasation at inflammatory sites. So, both immature and mature murine neutrophils are able to form hybrids *in vitro,* as well as *in vivo.* Regarding the required stimulus to acquire antigen presenting capacities of murine neutrophils, Radsak et al. reported the expression of MHC II and costimulatory molecules after coculturing of neutrophils with T cells and antigens [18]. Furthermore, Abi Abdallah et al. stressed the need for cell-cell-dependent contact between T cells and neutrophils, as separation by a transwell system abrogated the T cell-induced expression of MHC II on neutrophils [19]. These neutrophils are able to process and present antigens to CD4^+^ T cells, induce their proliferation and stimulate differentiation of T helper (Th) 1 and Th17 cells in vitro.

Resting human neutrophils under physiological conditions do not express markers typically found on APCs and are not able to induce proliferation of naive T cells. However, human peripheral blood neutrophils start to express DC markers (e.g., MHC II, CD80, CD83, CD86 and CC chemokine receptor 6) after in vitro activation by GM-CSF, interferon (IFN)-γ and interleukin (IL)-3. This generates neutrophil–DC hybrids with characteristics of APCs while maintaining typical features of neutrophils. It was shown that mature as well as immature human neutrophils are able to transdifferentiate into hybrids [13,20,21,22]. These APC-like neutrophils are not only an in vitro phenomenon, as they are also detected in patient samples (e.g., synovial fluid from patients with rheumatoid arthritis and blood from patients with Wegener’s granulomatosis), where the expression of MHC II, CD80, CD83 and CD86 on neutrophils is upregulated [13,22,23,24]. Vono et al. used freshly isolated human neutrophils to show their ability of presenting antigens in vivo to antigen-specific CD4^+^ T cells [14]. They found that the supernatant from antigen-specific, activated T cells induces the expression of MHC II and costimulatory molecules on neutrophils, which are then able to present antigens via interaction with T cells. Radsak et al. demonstrated that unstimulated human neutrophils are able to cause proliferation of antigen-specific T cells after co-culturing with T cells and their antigens [18]. This ability is dependent on MHC II, as blocking MHC II on neutrophils inhibits their antigen presenting capacity.

Lok et al. investigated the presence of neutrophils in LNs under physiological conditions [4]. Using flow cytometry and intravital imaging analysis, they were able to detect the presence of both human and murine neutrophils in LNs under homeostatic, unstimulated conditions. These cells enter the LNs via high endothelial venules and then circulate through the lymphatics. LN-neutrophils show a phenotype distinct from neutrophils derived from the peripheral blood. More specifically, LN-neutrophils display higher expression of CXCR4, MHC II and costimulatory molecules. They carry immunoglobulin G (IgG)-opsonized cargo, such as immune complexes, and are mostly found within the interfollicular region of the LNs, which is the area where CD4^+^ T cells are located for their antigen searches. Also, with the use of confocal microscopy, interactions are revealed between neutrophils and DCs in the LN cortex. Finally, when stimulated ex vivo with immune complexes, neutrophils upregulate MHC II and are able to activate CD4^+^ T cells. Altogether, this makes neutrophils ideal candidates to execute homeostatic immune surveillance, as well as to reactivate (autoreactive) T cells, by delivering circulating antigens (such as immune complexes or autoantigens) to the LN, where they can stimulate the adaptive immune cells.

## 2. Multiple Sclerosis (MS)

Multiple sclerosis (MS) is the most common immune-mediated inflammatory disease affecting the central nervous system (CNS), with an estimated prevalence of approximately 2.5 million people worldwide [26]. MS is a chronic, demyelinating, neurodegenerative disorder with no clear cause and still no cure available. About 70% of patients are diagnosed between the age of 20 and 50 years, with women having a 2- to 3-fold higher chance than men on being diagnosed with this disease. MS is a complex disorder caused by a combination of genetics (e.g., variants of the HLA-DR gene) and environmental factors, such as infectious agents, nutrition, with general effects on gut microbiota, and specific influences such as by vitamin D deficiency and lifestyle, including smoking [27,28]. All these elements together lead to post-translational modifications, alter the “self-proteome” and thus weaken the mechanisms of central and peripheral immune tolerance, thereby enhancing autoimmune responses against glycoprotein components of the neurite- and axon-insulating myelin sheath and oligodendrocytes in the CNS [26,29]. This altered immune response results in the pathophysiological characteristics of MS: inflammation, destruction of the myelin sheath, toxic effects on neurons and formation of multiple chronic inflammatory CNS lesions. Originally, these lesions were named “sclèrose en plaques” on autopsy, hence the name multiple sclerosis. The presence of brain versus spinal cord lesions and the spatial distribution of lesions varies between patients with different clinical subtypes of MS (*vide infra*) [30,31]. These typical features connect in an educated way to cause axonal injury, disrupt electrical signal conduction and yield a broad range of severe symptoms. MS implicates the loss of oligodendrocytes, the cells that generate myelin sheaths, which results in the destruction of the insulating layer around neurons [26]. Demyelination is accompanied by cellular infiltrates composed of both innate and adaptive immune cells. The following repair process, remyelination, is able to rebuild the myelin layer in the early phase of disease but this process becomes less effective over time. Finally, astrogliosis is important in generating a scar-like plaque. These inflammatory plaques can be macroscopically seen on autopsy and on magnetic resonance imaging (MRI), which is one of the techniques used to diagnose MS [32].

MS onset is most often diagnosed by the appearance of a clinically isolated syndrome of vision loss (optic neuritis) and gait disturbances, which are the first signs of the disease, reflecting inflammatory demyelination. These events occur either in focal or multifocal regions of the CNS, mainly in the optic nerve, brainstem or spinal cord. The majority of MS patients (85%) are characterized by a relapsing/remitting (RRMS) disease course, identified by episodes of symptoms alternated with recovery periods. Most of them will develop a progressive decline in function after several years to decades, called secondary progressive MS (SPMS) [33]. A small subset of patients (10%) experience a progressive disease course from the beginning without relapses, known as primary progressive MS (PPMS) [34]. The most common signs and symptoms of MS include autonomic, visual, motor and sensory problems, which vary considerably between patients depending on the location of MS lesions within the CNS. The following symptoms are most likely to occur: fatigue, cognitive impairment, muscle weakness, changes in sensation, blurred vision, coordination disturbances and bladder/bowel difficulties [33]. Cognitive impairment is experienced by 45–70% of MS patients and depends on duration of disease, age and disease subtype [35]. MS has a dramatic impact on the patient’s quality of life and well-being, resulting in many patients suffering from depression and having difficulties adapting their way of life within the society.

Current options of treatment target neuroinflammation but have only limited effect in reducing brain atrophy and established neurodegenerative damage. A real cure that stops disease progression is not available and drugs that are more effective display higher risks on severe adverse events. Due to the welcome increase in the amount of disease-modifying treatments (DMTs), costs keep rising and determination of a treatment strategy becomes complex [26,32,33,36].

The pathogenesis of MS remains incompletely elucidated, but the current mainstay is centered around adaptive immune cells as key players. Two models have been introduced to explain the detrimental immune response to CNS autoantigens [26]. The CNS intrinsic or ‘inside-out’ model proposes that the triggering immune event is situated within the CNS, which results in spreading of CNS antigens to the periphery where a proinflammatory environment provokes an autoimmune response against the CNS. Second, the ongoing CNS extrinsic or ‘outside-in’ model suggests that disease is initiated by peripheral activation of myelin-specific, autoreactive T cells [26]. The mechanism by which T cells are primed is believed to be molecular mimicry and/or bystander activation [29,37,38]. Molecular mimicry implies a shared immunological epitope between microbes and autoantigens from the host. Furthermore, bystander activation represents the activation of pre-existing autoreactive T cells through cytokines that were secreted as a result of a concomitant infection. Genetically susceptible individuals display defects in peripheral immune regulation which lowers the barrier for activation of autoreactive T cells, which occurs in the lymphoid organs [29,38]. Following their activation, peripheral T cells are able to pass the blood–brain barrier (BBB)/blood–spinal cord barrier (BSCB), choroid plexus or meninges and may enter the CNS under specific conditions where they become reactivated after encountering CNS-related autoantigens, presented by MHC II molecules on APCs [29,39,40,41]. The recent discovery of meningeal lymphatic vessels that link to both CNS immunosurveillance and neuroinflammation suggests that autoimmune T cells may also drain to the cervical LNs, contributing to epitope spreading to the periphery [42]. The activated T cells produce proinflammatory cues that induce an inflammatory cascade, resulting in the production of chemokines and cytokines and the further recruitment of immune cells to the site of inflammation. B cells are able to contribute to MS through production of autoantibodies or by acting as APCs [43]. In the brain, T cells, B cells, activated microglia and macrophages provoke myelin disruption, which induces the release of new CNS antigens to the periphery, called epitope spreading [44]. Altogether, this results in persistent inflammation, further injury to myelin and oligodendrocytes as well as axonal loss. The observed damage is probably caused in a direct manner via T cells, microglia, macrophages, complement and antibodies, as well as indirectly through the release of proinflammatory factors such as nitric oxide, matrix metalloproteinases (MMPs), tumor necrosis factor α (TNF-α) and IL-1β, [29,45,46,47], many of which are produced by neutrophils.

The autoantigens in MS that are responsible for the activation of T and B cells have yet to be defined. The autoantigenic peptides in MS may be generated by innate myeloid cells, as proposed in the REGA (remnant epitopes generate autoimmunity) model. This model was described about 25 years ago, when new cytokines and proteases of innate immunity were discovered and associated with MS. In this model, the emphasis was on innate rather than on adaptive immune mechanisms. In this way, cytokines, chemokines and extracellular proteases from myeloid cells, such as neutrophils, were placed in the spotlights as being critical molecules and cells in the initiation, maintenance and resolution of disease phases of MS and as phase-specific targets for the development of new therapies [29,48,49,50,51]. For the definition of an autoimmune disease, adaptive immune reactions in the form of autoantibodies or T cells armed with specific receptors against autoantigens are imperative [50]. Many studies have been conducted to prove this point, and the results are complex. Myelin proteins are considered to be suitable candidates for autoantigens but no definite proof is available that myelin is the unique target of autoreactive T cells [52]. Other studies propose autoantigens that do not reside within myelin, including αB crystallin and sperm-associated antigen 16 [53,54]. Planas et al. identified GDP-L-fucose synthase as a new candidate antigen for a subset of MS patients [55]. The GDP-L-fucose enzyme catalyzes a step in the fucosylation process of glycans. Both myelin proteins as well as myelin-oligodendrocyte glycoprotein (MOG) are covered with many fucosylated glycans. On one hand, this insight generates critical thinking about post-translational modifications by (i) intrinsic host and (ii) extrinsic environmental enzymes in microbiota and food [29,50]. On the other hand, a T cell response against the proposed antigen GDP-L-fucose synthase functionally generates a reduction in fucosylated glycans in the brain. This results in an altered immunogenic signal, possibly affecting neurons, and it may induce MS development via a proinflammatory response [55].

## 3. Evidence for the Implication of Neutrophils in MS and Its Animal Models

The involvement of innate immune cells, mostly macrophages and the brain-resident microglia, in MS pathology is increasingly recognized, as extensively reviewed elsewhere [41,56,57]. Recently, interest developed for the role of neutrophils in the pathogenesis of MS [16,58,59]. Being innate immune cells, these cells were previously overlooked in the context of MS autoimmunity for several reasons. First, neutrophils were previously supposed to be short-living cells without antigen-presenting capacity. Their short half-life and vulnerability discouraged most researchers from studying neutrophils in long-term animal models. For instance, the isolation and manipulation, even under the most careful conditions, may activate neutrophils and, therefore, passive transfer experiments, as often done with lymphocyte populations in mice, are extremely difficult to perform. Furthermore, neutrophils are not abundant in human CNS tissue in the context of MS. Cerebrospinal fluid (CSF) from MS patients was classically defined as clear fluid with only lymphocytes and other mononuclear cells (and oligoclonal immunoglobulin protein bands on electrophoretic analysis), in contrast with neutrophil pleocytosis in cases of bacterial meningitis. Our ignorance about neutrophils in MS may be explained by the fact that in post-mortem brain tissue, these cells are barely observed. It is almost impossible to demonstrate the transient or early-staged role of neutrophils. Another possibility is that the neutrophil phenotype is plastic, with expression of typical markers from both macrophages and DCs, as explained above. A last possibility would be that neutrophils contribute mostly to MS pathogenesis via affecting the periphery rather than the CNS [60].

In ongoing MS research, several mouse models have been established, among which are the experimental autoimmune encephalomyelitis (EAE) model and the cuprizone-induced encephalomyelitis model. EAE can be induced by active immunization of mice with a known myelin antigen (MOG, myelin basic protein (MBP) or proteolipid protein (PLP)) together with complete Freund’s adjuvant and *pertussis* toxin. Another method implies passive immunization, in which mice are immunized with pathogenic CD4^+^ T cells generated in donor animals [61,62]. This model is dependent on CD4^+^ T cells, antigen presentation and subsequent infiltration of T cells into the CNS, resulting in demyelination [61]. When mice are immunized with a sterile myelin suspension, containing glycoproteins, lipids and other macromolecules from mouse spinal cords, a (transient) hyperacute form of EAE develops with strong neutrophil involvement. In this condition, neutrophils are even detected in the CSF [63]. In the cuprizone model, demyelination is induced by feeding mice with cuprizone, a neurotoxic copper chelator. This results in oxidative/nitrative stress leading to oligodendrocyte loss and demyelination with spontaneous remyelination after cessation of supplementing the food with cuprizone [64]. The cuprizone model is believed to largely depend on microglial-mediated demyelination and other CNS-centric mechanisms without evident contribution of antigen presentation and T cells [65]. Both models are shown to have merits and are indispensable for studying different aspects of MS, yet do not fully reproduce the complete human disease. Indeed, the EAE model is a CD4^+^ T cell-driven disease model and does not take into account the importance of other immune cell subtypes. The cuprizone model is excellent to study de- and re-myelination but neglects the involvement of the autoimmune T or B cells [66].

After the description of the hyperacute EAE model [63], further evidence for the involvement of neutrophils in the pathogenesis of EAE was demonstrated. For instance, neutrophil numbers are increased in both the periphery and CNS before and during the onset of clinical EAE with mouse spinal cord homogenates [67,68]. Neutrophils had already accumulated at the meninges in EAE mice after 24 h of disease induction and their numbers increased during preclinical and peak stage [69]. Furthermore, a reduced severity and delayed onset of EAE is seen when neutrophils are depleted prior to but not after disease initiation [70,71]. This might indicate that neutrophil function is mostly important for disease onset and the initiation of relapses. Neutrophil infiltration in EAE can be linked to the expression of granulocyte-colony stimulating factor (G-CSF), which is increased prior to disease onset and correlates with the number of infiltrating neutrophils [72]. Also, *G-*CSF receptor (*Csf3r*) knockout mice display lower numbers of neutrophils in their circulation and show resistance to EAE development. The recruitment of neutrophils to brain and spinal cord in EAE depends on the glutamic acid-leucine-arginine-positive (ELR^+^) chemokines (CXCL1, CXCL2 and CXCL6), produced by Th17 cells that activate CXCR2. Mice with induced CXCL1 expression in astrocytes show an increased EAE severity that correlates with a higher number of recruited neutrophils to the CNS [73]. Infiltration of other immune cells, such as T cells and macrophages, is not found to be increased after CXCL1 expression, indicating that these cells do not explain the enhanced disease severity. Blocking of both CXCL1 and CXCR2 in mice results in a delayed onset of EAE, combined with a lower number of neutrophils and reduced EAE symptoms [71,73]. Furthermore, transfer of CXCR2^+^ neutrophils into *Cxcr2* knockout mice re-establishes their susceptibility to EAE. In the cuprizone model, it is demonstrated that CXCR2^+^ neutrophils are both necessary and sufficient to induce demyelination [74].

In the blood of MS patients, increased concentrations of neutrophil-activating chemokines and neutrophil-derived enzymes are detected (e.g., CXCL1, CXCL8, NE, MPO) and these molecules are associated with the formation of new inflammatory lesions [72,75,76]. The neutrophil-to-lymphocyte ratio is proposed as a marker for disease activity, as it was elevated in MS patients and higher in patients experiencing relapse compared to remission [77]. Also, neutrophils isolated from RRMS patients’ blood are found to be more primed, express more inflammatory markers and display resistance to apoptosis [76,78,79]. Neutrophils have further been detected in the CSF of MS patients at early disease stages and at the beginning of a relapse phase, suggesting their active involvement in disease [80]. This phenomenon was no longer observed in patients with long disease duration, pointing at a more prominent role for neutrophils in the onset of disease or relapse. Additionally, the neutrophil cytosolic factor 4 (*NCF4*) gene, encoding one of the subunits of the nicotinamide adenine dinucleotide phosphate complex in neutrophils, was identified as a genetic factor predisposing to MS in the most recent genome-wide association study of the international MS consortium [81].

## 4. Neutrophil Effector Functions and Their Link to MS Pathogenesis

Neutrophils exhibit a broad range of effector functions that may be relevant for the pathogenesis of immune-mediated diseases such as MS (Figure 2).

### 4.1. Phagocytosis

Phagocytosis is a well-established function of neutrophils, which is used to defend the host against invading pathogens but also to eliminate cellular debris [82]. Demyelination in EAE and MS is the result of an inflammatory cascade in the CNS, ending with the phagocytosis of oligodendrocytes and myelin by microglia and myeloid cells [83,84]. Until recently, it was believed that macrophages and their related microglia are the most important myeloid cells responsible for phagocytosis and digestion of the myelin layer in MS. However, after the publication of an image of a mouse EAE neutrophil engulfing a piece of myelin, the possibility of neutrophils being complementary to macrophages in the phagocytosis of myelin was proposed [85]. It was shown previously in the cuprizone model for MS that accumulation of myelin debris impairs remyelination, whereas clearance enhances it [86]. In this respect, a relevant question remains unanswered: is phagocytosis of myelin a proinflammatory mechanism that contributes to disease pathogenesis or is it a pro-resolving attempt to attenuate the ongoing immune reaction?

### 4.2. Release of Inflammatory Mediators

#### 4.2.1. Interleukin-1β (IL-1β)

It has been known for a long time that the IL-1 system plays a key role in CNS neuroinflammation and repair. IL-1β is mainly produced by neutrophils and monocyte-derived macrophages (MDMs) as a pro-peptide that needs to be cleaved to become fully active. Whereas MDMs synthesize IL-1β *de novo* after appropriate stimulation [87], for instance by the ligands of Toll-like receptors, neutrophils contain this interleukin prepacked in their secretory vesicles and ready for immediate release after triggering [88,89]. Furthermore, neutrophils contribute to the maturation of pro-IL-1β through the secretion of serine proteases and metalloproteinases [90]. Allen et al. showed that IL-1β causes activation and recruitment of neutrophils in vivo and that migration across IL-1β-stimulated brain endothelium generates a neurotoxic phenotype of neutrophils which induces death of cultured neurons in mice [91]. Interest is rising for a role of IL-1β in MS where it seems to play a dual role [92]. IL-1β is present in CSF and brain lesions of MS patients and a high IL-1β/IL-1 receptor antagonist ratio is linked to an increased risk on developing RRMS. In the EAE model for MS, it is found that neutrophils and MDMs produce pro-IL-1β concomitantly with their migration over the BBB. This aggravates EAE via activation of the IL-1 receptor on endothelial cells, leading to the secretion of proinflammatory cytokines/chemokines (GM-CSF, G-CSF, IL-6, CXCL1/2, etc.), resulting in more recruitment and activation of myeloid cells. GM-CSF production induces a positive feedback loop by stimulating the IL-1β production and causes the differentiation of MDMs into professional APCs, thereby stimulating autoreactive T cells [92,93]. As GM-CSF is one of the factors believed to trigger formation of neutrophil-DC hybrids, the feedback loop with IL-1β might stimulate hybrid cell formation (*vide supra)*. Mice deficient in IL-1β, but not IL-1α signaling, display an attenuated EAE disease severity [94]. Additionally, several DMTs approved for MS lower the amounts or the maturation of IL-1β [95,96]. Aside these demonstrations of detrimental roles of IL-1β, CNS remyelination in rodents is also dependent on IL-1 signaling [97], stressing the dual aspects for the IL-1 family members in MS.

Another aspect to highlight about the role of IL-1β in MS and EAE is a positive feedback loop with IL-17A. The differentiation from naive CD4^+^ T cells into proinflammatory Th17 cells is stimulated by several cytokines, including IL-1β, which also induces secretion of IL-17A from these Th17 cells [98]. The effect of IL-17A in autoimmunity includes stimulation of the production of proinflammatory cytokines, MMPs and chemokines/glycoproteins to recruit neutrophils such as CXCL1, CXCL2, C-C motif chemokine ligand 2 and G-CSF [99]. In EAE, production of IL-17A induces chemokines to trigger the influx of IL-1β-producing neutrophils into draining LNs. The recruited neutrophils are responsible for the priming of pathogenic Th17 cells in the LN, which enhances CNS autoimmunity. Additionally, *Il-17a^−/−^* mice, which are less susceptible to EAE disease and show reduced IL-1β production in their LNs, re-establish their susceptibility to EAE after treatment with IL-1β. This IL-1β is responsible for the expansion of the Th17 population in EAE and creates a positive feedback loop by enhancing IL-17A expression [98].

#### 4.2.2. Matrix Metalloproteinases (MMPs)

MMPs, endopeptidases belonging to the metzincin superfamily, play an important role in a variety of neurological diseases, as well as in neurophysiological conditions, the latter by regulating synaptic plasticity and neuro-regeneration [100]. Up to now, 24 different mammalian MMPs have been discovered with each of these having their own specific substrates. MMPs are able to degrade components of the extracellular matrix, cleave adhesion molecules or receptors, and in this way, contribute to the inflammatory cascade [101]. They are produced as inactive pro-MMPs that are activated after cleavage by various proteases, other MMPs or via oxidation by ROS [102]. One function of MMPs is to regulate chemotactic gradients by cleavage of cytokines/chemokines, thereby altering the chemoattractant properties of the latter [103]. Whereas in most cases MMPs and other proteases decrease the activity of chemokines, in neutrophil biology, a notorious exception has been documented. Indeed, proMMP-9 released from neutrophils and activated into MMP-9 by ROS (and other proteases) has the capacity to truncate human IL-8/CXCL8 into a ten-fold more active chemokine [104]. Many additional studies and roles of MMP-9 in mouse EAE are mentioned in a recent publication by Ugarte-Berzal et al. [105]. MMP-2 and MMP-9 synergize in the attraction of neutrophils to the site of injury by enhancing the potency of human CXCL8 and mouse CXCL6 after cleavage. In seminal work, the expression of MMP-9 in situ in human MS and mouse EAE has been recently evidenced with the use of novel probes and high-end imaging techniques. MMP-9 activity is found to be increased during MS and EAE disease episodes and decreased during recovery phases [106]. Another function of MMPs, which was discovered about 30 years ago [63], is their contribution to BBB leakage. More specifically, MMP-2 and MMP-9 degrade tight junctional and basement membrane proteins, which leads to BBB disruption and promotes infiltration of leukocytes, including neutrophils, into the CNS. This was also supported by animal studies, in which *Mmp2^−/−^* and *Mmp9^−/−^* double knockout mice are completely resistant to EAE development [107]. It was found that mice treated with MMP-9 inhibitors show a decrease in their BBB permeability [108]. In MS patients, an increase in MMP-9 levels was measured in both CSF and blood samples, and these levels correlate with disease activity [109,110,111]. Inhibitors of MMPs may therefore be useful in the clinic to reduce neuroinflammation in MS. For instance, tetracyclines were found to inhibit MMP-9, with minocycline and doxycycline as the most potent drugs [112]. Thanks to major efforts by Canadian MS researchers, minocycline as an inexpensive drug has recently been brought to the MS clinic with comparable beneficial effects to the much more expensive interferon-β [113].

#### 4.2.3. Myeloperoxidase (MPO)

MPO is a peroxidase enzyme stored in the azurophilic granules of neutrophils and can be released into the extracellular space during respiratory burst, thereby generating toxic radicals and oxidants to kill pathogens [8]. Neutrophils are affected by MPO in different ways: (1) MPO is able to activate neutrophils, (2) MPO can stimulate the accumulation of neutrophils in the CNS and (3) MPO causes a delay in the apoptosis of neutrophils [114,115,116]. The contribution of MPO to MS pathogenesis was evidenced by an increase in MPO levels in the plasma and white matter of patients [75]. One of the mechanisms by which MPO influences EAE disease pathogenesis is believed to be the increase of BBB permeability, as shown by Zhang et al. [117]. Up to now, no MPO inhibitor was used in clinical studies, due to a high level of toxicity and low degree of specificity. Zhang et al. developed a new, specific, non-toxic MPO inhibitor, N-acetyl lysyltyrosylcysteine amide (KYC) [117]. When tested in EAE, these authors detected that inhibition of MPO at the peak of disease resulted in a reduction of myeloid cell infiltration and disease severity, concomitantly with an increase in BBB integrity and oligodendrocyte regeneration [118]. This indicates that MPO exhibits its pathogenic role after disease initiation, prior to the peak of disease. The mechanism by which MPO exerts its effect in EAE/MS is not completely elucidated.

### 4.3. Breakdown of the Blood–Brain Barrier (BBB)

Under physiological conditions, immune cells are restricted from entering the CNS due to the presence of the BBB and the BSCB. The major histological principle of the BBB is the presence of a second parenchymal basement membrane (produced by the astrocyte endfeet), aside the primary endothelial basement membrane [119]. During neuroinflammation, the BBB is temporarily and locally altered, opening the gates for the influx of leukocytes. These cells pass through the endothelial basement membrane, after which they remain trapped in the perivascular space, lined with the second parenchymal basement membrane. This phenomenon of local leukocyte trapping is the so-called ‘perivascular cuffing’. Leukocytes in EAE accumulate for several days in the perivascular space where they may become reactivated through proinflammatory cytokines or by antigen presentation. This may lead to local protease production and disintegration of the parenchymal basement membrane. In this way, the leukocyte can enter into the CNS parenchyma [120]. Disease symptoms only develop after this last step of leukocyte transmigration through the parenchymal basement membrane, indicating that this a crucial step in disease development and a target for therapy. It is believed that neutrophils contribute to breakdown of the BBB and to the final translocation of leukocytes into the brain parenchyma. In EAE, disruption of the BBB is observed concomitantly with the appearance of neutrophils in the CNS, whereas depletion of neutrophils in the circulation causes an increase in BBB integrity. By confocal imaging, Aubé et al. localized neutrophils in active MS lesions at sites where the BBB is leaky [47]. Additionally, neutrophil depletion is shown to decrease cellular infiltration and results in an increased perivascular cuffing in EAE [10]. Also, neutrophils from EAE mice are found to migrate more efficiently over an in vitro BBB cell culture model compared to healthy control neutrophils [47]. It has been hypothesized that neutrophils could mediate BBB breakdown via contact-dependent mechanisms and through the secretion of enzymes (MPO), proteases (MMPs) and free radicals (ROS) [32].

BBB breakdown has been associated with various neurological conditions, such as stroke, MS and Alzheimer’s disease, which all share the feature ‘oxidative stress’. This is an imbalance in the oxidant/antioxidant ratio, ending with overweight at the side of oxidants, as also seen in MS patients [121,122]. In active demyelinating MS lesions, oxidative damage to various proteins, lipids and nucleotides was detected [123]. Peripheral blood neutrophils from MS patients are found to be more primed, accompanied by an enhanced oxidative burst [76]. Transmigration of neutrophils across an activated endothelium increases their ROS production and degranulation capacity [124]. ROS production is directly involved in demyelination as well as damage to astrocytes and axons in EAE and MS [125]. ROS are also critical molecules for the activation of MMP enzymes [126], which in turn degrade the endothelial basement membrane, resulting in gaps in the BBB. More specifically, MMP-2 and MMP-9, together with cytokines and chemokines, cooperate at the parenchymal side of the BBB to regulate breakdown of the basement membrane and subsequent leukocyte infiltration in MS [119]. MMPs, produced by neutrophils and macrophages, are able to cleave dystroglycan. This is a transmembrane receptor attaching the astrocytic endfeet to the parenchymal basement membrane of the BBB. Double *Mmp2^−/−^/Mmp9^−/−^* knockout mice show resistance to EAE, with an inhibited cleavage of dystroglycan and a blocked leukocyte penetration into the CNS [120]. Gerwien et al. imaged MMP activity in EAE and MS brain to monitor lesion formation in the tissue, and proposed that MMP activity is ahead of BBB breakdown and subsequent lesion formation [106]. They measured MMP activity, which is limited to sites of leukocyte influx at the parenchymal border of the BBB and is detected concomitantly with MMP-producing CD45^+^ leukocytes. These authors proposed MMP-9 as a suitable marker in EAE for leukocyte penetration at the parenchymal side of the BBB, whereas imaging of activated MMPs precedes MRI-detectable lesion formation in the CNS.

The MPO enzyme produced by neutrophils converts hydrogen peroxide into reactive secondary oxidants, which is part of the so-called “respiratory burst” and creates a cytotoxic environment against invading pathogens [127]. Using an MPO inhibitor, Zhang et al. demonstrate the importance of MPO on BBB breakdown in the EAE model [117]. After five days of MPO inhibition, a completely restored BBB integrity was observed, concomitantly with a decrease in EAE disease severity and a reduced absolute number of neutrophils. Additionally, Üllen et al. used a lipopolysaccharide-induced model of neuroinflammation with MPO-deficient mice to demonstrate that the induced BBB leakage was lower in *Mpo^−/−^* mice compared to wild-type animals [128].

The migration of neutrophils through the BBB induces the secretion of IL-1β, which in turn triggers GM-CSF production by endothelial cells and T cells. GM-CSF increases the IL-1β secretion, creating a positive feedback loop that enhances the neuroinflammatory cascade in EAE [97]. An important role for the Th17-produced chemokines (CXCL1/2) and CXCR2 pathway in BBB breakdown in EAE is evidenced by Carlson et al., who demonstrate that blocking CXCR2 inhibits BBB breakdown and immune cell infiltration. The same effect on BBB integrity is obtained when neutrophils are depleted from the circulation and the injection of CXCR2^+^ neutrophils recovers EAE susceptibility in *Cxcr2^−/−^* mice [71].

### 4.4. Production of Neutrophil Extracellular Traps (NETs)

A recently discovered function of neutrophils is the formation of NETs, sometimes accompanied by a necrosis-like, programmed form of cell death called NETosis [9]. To do so, neutrophils release DNA, of which the majority has mitochondrial origin (mtDNA). This DNA is able to bind histones and antimicrobial products, such as cytotoxic granulocytic proteins and neutrophilic enzymes (NE, MPO, MMPs). This mtDNA is differently methylated than nuclear DNA and functions as an endogenous danger-associated molecular pattern and as a ligand for DNA-sensing pattern recognition receptors to potentiate the immune response [129]. NETs are formed to trap and kill microorganisms, but they also cause activation of DCs and priming of T cells. NET formation and NETosis were found to be dependent on ROS and ATP production in humans [130]. A disturbed balance of NET formation and clearance has been associated with a variety of autoimmune diseases, such as psoriasis and systemic lupus erythematosus [131,132].

NETs are elevated in the serum of MS patients compared to healthy controls, probably due to the chronic inflammatory environment that is responsible for priming of neutrophils [76]. Circulating NETs were found to be abundant in a subset of patients with RRMS and were significantly elevated in male patients, who suffer mostly from a worse prognosis, compared to female patients [76,133]. This might contribute to gender-dependent molecular differences in mechanisms of disease, which is also highlighted in a transcriptomics study [134]. Recent studies in MS suggest the possibility that NETs would have a cytotoxic effect on the BBB and induce injury of adjacent neurons and other cells of the CNS [133]. This is supported by EAE studies, showing that depletion of NET-associated proteins (MPO, NE) decreases disease severity and increased BBB integrity [117,118]. Allen et al. showed that transmigration of murine neutrophils through an activated cerebrovascular endothelium induces a proinflammatory, neurotoxic phenotype that subsequently leads to NET release [91]. However, real proof that NETs contribute to BBB breakdown in MS still needs to be provided. Another role attributed to NETs is the cleavage of circulating immune complexes in MS, as proposed by Paryzhak et al. [135]. They highlighted the potential of NET-related proteases to modify circulating IgG immune complexes, leading to the unfolding of internal glycoepitopes.

### 4.5. Autoantigen Generation and Presentation

#### 4.5.1. Autoantigen Generation: Post-Translational Modifications

(a)Proteolysis—Remnant epitopes generate autoimmunity (REGA model)

With MS being an immune-mediated disease, autoantibodies and autoreactive T cells are present and are directed against autoantigens, among which are myelin proteins. An original model for the generation of these autoantigens and the associated autoreactive T cells is the REGA model. In this model, which was re-iteratively perfected into a paradigm of autoimmunity, a certain inflammatory trigger initiates the production of cytokines and chemokines (such as IL-8/CXCL8) that attract myeloid cells to the site of inflammation [50]. Here, activated neutrophils secrete their pre-stored granules, containing proteases such as MMP-9, after which these enzymes become fully active. Subsequently, myelin proteins are proteolytically cleaved into remnant epitopes, leading to the presentation of these remnant antigens by APCs in MHC context to autoreactive T cells [48]. It was shown in EAE that MBP epitopes are proteolytically generated by MMP-9 in the initiation phase and these MBP peptides activate T cells [136,137]. In addition, although MMP-9 does not cleave IgG, it is able to cleave MBP from circulating immune complexes, thereby degrading the antigen from immune complexes in the remission phase and thus playing a protective role in the remission phase by promoting clearance of the autoantigen [50,105]. Another example of a remnant epitope in MS is the autoantigen αB-crystallin, which is a heat shock protein found in active MS lesions for presentation to T cells, but its role in MS is still unclear [138,139]. It is believed that the intact αB-crystallin protein is protective for MS, as knockout mice showed more severe EAE [140]. However, and perfectly in line with the REGA paradigm, the MMP-9-generated αB-crystallin epitopes provoke murine T cell proliferation and contribute to autoimmunity [141].

(b)Citrullination

Citrullination is defined as the enzymatic conversion of an arginine residue into a citrulline and is catalyzed by the protein arginine deiminase (PAD) enzyme family with 5 isozymes in humans. Overexpression and increased enzyme activity of PAD has been associated with MS [142]. Both PAD2 and PAD4 are found in neutrophils, and PAD4 induces histone citrullination, which is essential for the chromatin decondensation step that precedes NET formation in humans and mice. In the CNS, PAD2 is normally the most abundant form, but PAD4 is found to be overexpressed and activated in MS brains and its animal models [143,144]. MBP was found to be citrullinated by PAD2 and could be isolated from MS brains, and the degree of MBP-citrullination was correlated with disease severity [29,145]. This deimination causes loss of positive charge and results in an open conformation of MBP, rendering it more susceptible to proteases and generation of remnant epitopes [144]. Also, the citrullinated MBP protein induces fragmentation of lipid vesicles of the myelin layer, which can contribute to demyelination in MS [146,147].

#### 4.5.2. Autoantigen Presentation

Neutrophils exert an immunomodulatory function for adaptive immune cells by acting either indirectly, through maturation of APCs, or by directly affecting T cells. In the pathogenesis of MS and its mouse model EAE, a crucial step is the encounter of CNS-infiltrating T cells with cognate APCs to reactivate these autoreactive T cells. The proinflammatory environment for reactivation of T cells is created by different cell types, such as CNS-resident microglia as well as infiltrating DCs and macrophages [85]. The classical processes of antigen presentation start with the uptake of extracellular proteins that are intracellularly processed to short peptides within the APCs by the proteasome or by lysosomal enzymes and presented on, e.g., MHC II molecules. Next, CD4^+^ T cells are activated at the immunological synapse, which is formed by physical contact between T cells and APCs [148]. Neutrophils were found to play an essential role in this process, as they can contribute to the maturation of microglia and infiltrating monocytes, whose capacity to reactivate T cells subsequently increases [10]. It was demonstrated that CNS-derived neutrophils and their supernatant, but not bone marrow-derived neutrophils from EAE mice are able to induce MHC II and CD80/86 expression on DCs in vitro. Additionally, after in vivo depletion of neutrophils, APC-maturation and recruitment of DCs to the CNS was impaired [10]. Finally, neutrophils are found to promote the antigen presenting capacity and meningeal accumulation of B cells in EAE [149].

Recent literature indicates a role for human and murine neutrophils themselves as alternative APCs in autoimmune diseases for both T cells in the interfollicular region of LNs and for B cells in the spleen [25,150]. Whereas DCs are supposed to act as the major APCs in EAE, recently it was evidenced that DCs can be replaced by alternative APCs and are not strictly necessary to elicit T cell responses [150]. One possibility might be that neutrophils are able to replace DCs as alternative APCs in EAE/MS, hereby supporting epitope spreading and influencing adaptive immunity. Recent functional studies suggest that murine neutrophils are able to express both MHC II and costimulatory molecules and are also capable of processing and presenting antigens to activated T cells, depending on cellular contact [151]. The possibility arises that neutrophils exhibit a pathological role in human autoimmune diseases through the transport of antigens to LNs, presentation of antigens to T cells and interaction with APCs [14]. Also, by differentiating into APC-like cells, neutrophils are able to escape from apoptosis. By doing so, neutrophils delay the inflammatory resolution and contribute to the evolution of inflammation to chronicity. Human neutrophils under physiological circumstances contain cytoplasmic granules with CD80/86, and a fraction of these neutrophils (10%) carry granules with MHC II. Following neutrophil activation, secretory vesicles and granules translocate to the cell surface. MHC II can also be synthesized de novo in neutrophils, after stimulation with cytokines such as IFN-γ [152].

Both human and murine neutrophils develop the capacity to internalize, process and cross-present exogenous antigens in MHC I context to CD8^+^ T cells, which they acquire during coculturing with CD8^+^ T cells or the cytokines they produce [18,153,154]. Beauvillain et al. showed that murine peritoneal- or bone marrow-derived neutrophils are able to cross-present antigens both in vitro and in vivo to CD8^+^ T cells equally efficiently as macrophages [154]. Additionally, these murine neutrophils are also able to prime naive CD8^+^ T cells in vitro and in vivo after pulsation with antigen. As activated CD8^+^ T cells are implicated in MS pathogenesis, it can be hypothesized that neutrophils might be able to cross-present autoantigens, thereby contributing to autoreactive CD8^+^ T cell responses. Circulating neutrophils from patients with acute sepsis are primed and characterized by a prolonged survival, which is induced by unconventional T cells. These neutrophils are able to prime both CD4^+^ and CD8^+^ T cells and display APC-like properties, caused by unconventional T cell-secreted cytokines. Additionally, these neutrophils acquire the ability to cross-present exogenous antigens to CD8^+^ T cells [155].

## 5. Proinflammatory versus Pro-Resolving Neutrophils in EAE and MS

Besides all proinflammatory functions of neutrophils that contribute to the onset and further disease course of MS, neutrophils and their products also exhibit pro-resolving roles. Neutrophils are able to actively contribute to the resolution of inflammation through a couple of mechanisms. First, they are able to produce pro-resolving lipid mediators such as resolvins, protectins and lipoxins [156]. These mediators inhibit neutrophil infiltration and strengthen the uptake of apoptotic neutrophils by macrophages. Second, neutrophils scavenge inflammatory chemokines and cytokines through the use of decoy and scavenger receptors [157]. Finally, efferocytosis of apoptotic neutrophils by macrophages polarizes the latter to the M2-like phenotype and negatively regulates inflammation [158].

Zehnter et al. isolated neutrophils from the CNS of IFN-γ-deficient EAE mice, which had a markedly increased number of infiltrated neutrophils and showed exacerbated disease compared to normal EAE animals [159]. The isolated neutrophils from both groups of EAE mice are strong suppressors of T cell immune responses and this suppressive function requires T cell-derived IFN-γ. The inhibitory activity of neutrophils on autoimmune T cell actions depends on the production of nitric oxide synthase, which is induced by IFN-γ. This may explain the increased disease scores and fatality of EAE in *Ifng* knockout mice, where higher numbers of neutrophils are recruited. Indeed, IFN-γ normally inhibits the IL-17-induced ELR+ chemokines that recruit neutrophils. Also, in these knockout animals, nitric oxide synthase is not induced, and this effect subsequently diminishes the immunosuppressive activity of neutrophils on autoreactive T cells. In a recent study, using intravital imaging, Wasser et al. revealed that myeloid cells (microglia and macrophages were studied) are able to actively phagocytose CNS-invading Th17 cells [160]. This results in cell death of the engulfed cells and an attenuation of clinical EAE severity. This indicates that CNS-resident and CNS-invading myeloid cells may use their phagocytotic function in attempts to engulf potentially pathogenic Th17 cells as a first-line defense mechanism. Whether this property is shared by neutrophils remains to be investigated. Finally, Haschka et al. immunophenotyped samples from patients with different types of MS. In patients with inactive RRMS, an expansion in the neutrophil subset is observed together with a decrease of the lymphocyte compartment, which might hint towards a regulatory role for neutrophils [161].

## 6. Conclusions

Here, we summarized the current and increasing knowledge about neutrophils and their effector functions as contributing elements in the pathogenesis of MS and its animal models. Whether neutrophils exhibit an exclusive proinflammatory role in the onset and progression of disease or also aid in the resolution of inflammation remains to be elucidated. These topics are gradually gaining interest, because these may have implications for diagnosis and treatment of MS. Both normal microbiota as well as a number of infections have been associated with MS disease onset, possibly also as primary triggers of relapses. We suggest that neutrophils as primary innate defense cells against bacteria and other micro-organisms may understandably be implicated in the onset or every new early phase of MS. Experimentally induced episodes of inflammation in mouse EAE models and of demyelination in cuprizone mouse models for MS illustrate the involvement of neutrophils. The casual observation of neutrophils in MS lesions and the established detection of neutrophil-related molecules in plasma or CSF of patients with MS strengthen our view that further research in this area is relevant to define better biomarkers for MS diagnosis and prognosis. The connection between currently used DMTs and neutrophils differs greatly between drugs, making it presently hard to study possible effects of neutrophils on treatment success or observed adverse effects. IFN-β, the first DMT used for MS-treatment, decreases the number of circulating neutrophils in RRMS patients, further supported by a decrease in neutrophil infiltration in animal models, probably by downregulation of CXCR2 ligands [162,163]. The small-molecule immunomodulator fingolimod provokes apoptosis of cultured human neutrophils and elicits an inhibitory effect on neutrophils in animal models [164,165]. The immune-suppressive dimethylfumarate impairs neutrophil function in vitro and in mouse models [166]. Concerning immunomodulatory antibodies, natalizumab has no effect on neutrophil function or number, whereas alemtuzumab induces neutrophil depletion and impairs neutrophil function [167,168].

In addition, neutrophils and their phenotypes may be linked to specific disease courses of MS. For instance, the observation of an expansion of CD15^+^ neutrophils in inactive RRMS may be helpful for early detection and/or determination of the response to treatment [161]. During the remission phase however, granulocyte numbers were decreased in RRMS patients [169]. Also, neutrophil-targeting therapies or treatments that interfere with neutrophil effector functions represent a new option for testing in MS models. An example of this is related to the biology of MMP-9, a major neutrophil protease and detrimental factor in MS and EAE onset. The endogenous antagonist of MMP-9 is a tissue inhibitor of metalloprotease-1 (TIMP-1), and this molecule is not produced by neutrophils [170]. With recent research, it was established that the endogenous induction of TIMP-1 with the use of oncostatin M results in strong remyelination in the cuprizone animal model [171]. This study, as well as other approaches to enhance remyelination [172], illustrates that new approaches for treatment of MS may originate from the in-depth dissection of known and unexpected cellular and molecular actions. We are convinced that further development of neutrophil research in MS will yield interesting novel insights.

## Figures and Tables

**Figure 1 ijms-21-04558-f001:**
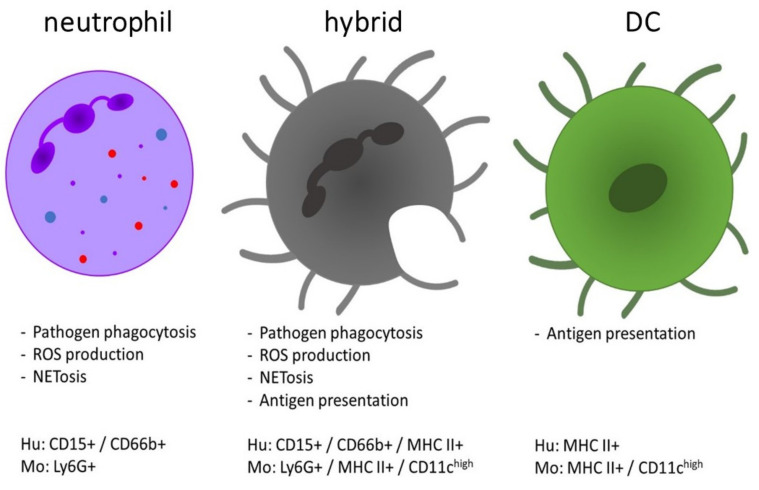
Comparison of the functions and phenotype of neutrophils, dendritic cells (DCs) and the neutrophil-DC hybrids. The hybrid population is characterized by combined functions of both DCs (antigen presentation and T cell activation) and neutrophils (phagocytic clearance of pathogens, etc.). The expression profile of hybrid cells is defined as a mixed phenotype with expression of markers from both neutrophils and DCs [10,11]. They morphologically resemble DCs, whereas their nucleus was detected as both an oval shape or as multi-lobular [25]. Hu = human, Mo = mouse.

**Figure 2 ijms-21-04558-f002:**
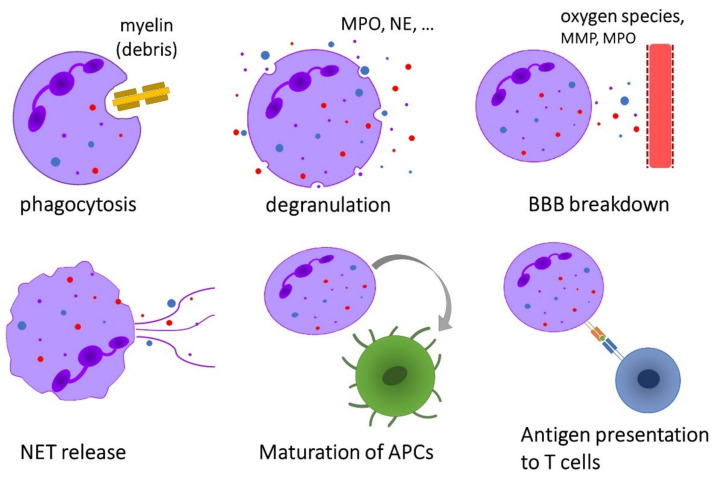
Overview of the effector functions of neutrophils that are possibly implicated in the pathogenesis of multiple sclerosis.

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
