# Peer review of "Neutrophils: Underestimated Players in the Pathogenesis of Multiple Sclerosis (MS)"

_ijms, 2020, doi:10.3390/ijms21124558_

Round 1

Reviewer 1 Report

Summary:
In this review, the Authors aimed to provide an overview of the possible role of neutrophils in the pathophysiology of multiple sclerosis (MS).

These are my comments.

- Research strategies and selection criteria for the different papers discussed should be clearly defined.

- Line 144. reference number 27 does not seem very appropriate here.

- Lines 149-150. There is no clear evidence that alcohol and other forms of abuse are associated with an increased MS risk.

- As written, the pathophysiology of MS is quite vague and simplistic. A better overview should be provided.

- Lines 158-159. Can the Author provide a reference for the statement regarding the more severe involvement of the brain compared to the spinal cord? Both the brain and spinal cord are frequently affected in MS.

- Lines 173-174. At present, the concept of benign MS is very debated. I would skip this concept. Several definitions have been proposed for benign MS but they are mainly related to a low EDSS score. However other aspects, such as cognitive impairment are not well considered.

- In 2014 a revision of the clinical phenotypes has been proposed by Lublin et al., (Neurology 2014).

- Recently, the role of B cells has been reconsidered. I think that the contribution of B cells should be better discussed in the review.

- Lines 232-233. Also, other recent studies and reviews have consistently demonstrated the role of microglia in MS.

- Neutrophil functions should follow the same order in the figure and in the text. Phagocytosis is presented as first in the figure, but as second in the text.

- Sometimes is difficult to understand whether the Authors referred to human or animal studies. Please clarify better what the Authors were referring to.

- Are there more findings in humans regarding the role of neutrophils in causing/promoting MS? The pathological processes of the experimental models of MS are quite different from what happens in MS, thus what found in these experiments could not really reflect what happens in vivo in humans. For instance, experimental models are based on acute inflammation and demyelination, but not in chronic processes including also neurodegeneration. These aspects should be clearly discussed.

- Some relevant aspects were not discussed in the review. For instance, possible differences among different clinical phenotypes and phases of the disease were not mentioned.

- Which could be the effects (if any) of the different MS-specific disease-modifying therapies on neutrophils?

- The review is mainly focused on the possible effects of neutrophils as a trigger of inflammatory demyelinating activity, but not as a possible factor involved in disease onset. Is there any evidence also for their role in MS etiopathogenesis? Please clarify these aspects.

Reviewer 2 Report

This is a very interesting, well written and exhaustive review on the actual understanding the detrimental mechanisms exerted by neutrophils in MS and EAE, its animal model.

I would have only a few considerations:

Major points.

Line 16, 295 and 486. The autoimmune nature of multiple sclerosis would need more cautious wording. As well discussed by authors (lines 217-230), it is generally accepted that MS is an inflammatory disease but not that it is an autoimmune disease. The involvement of autoreactive T cells and autoantibodies in lesion pathogenesis in MS are rather controversial.

Lines 187-189 The “peripheral immune activation in MS” description, must be implemented by the recent discovery of the cerebral lymphatic system, particularly the lymphatic vessels draining tissue debris during neuro-pathologies as also as epitope spreading mechanism.

Line 196-198. The BBB breakdown is now not considered the main route of entrance for pioneer peripheral immune cells CNS. New more details acquired on peripheral cell trafficking into CNS suggest preferentially earliest leukocyte infiltration across choroid plexus and the blood-cerebrospinal fluid barrier.

Minor

Line 99. “neut17ophils” must be corrected.

Line 438. “autors” must be corrected.

Lines 608-610. Please, provide the related references.

Reviewer 3 Report

The authors provide a well-written manuscript considering both English and clarity of the content. The manuscript generally is well organized and easy to follow, even though an enriched description of neutrophils (in general) in the first section may add more value to the review. All the aspects taken in consideration are very interesting, but there are some small imperfections that need to be revised. Overall, the review is pleasant, mentioning an important topic in the field of MS research.

Major comments

In the third paragraph, authors focused on different animal model of MS, in particular the experimental autoimmune encephalomyelitis (EAE) and the cuprizone-induced encephalomyelitis (CIE). It is known that EAE can be induced by using different strains and antigens, both by active immunization with antigens and passively through injection of encephalitogenic T cells, leading to distinct forms of disease. It is suggested to slightly pause on this, elucidating also some limits in the use of this animal models, see for example Constantinescu C.S. et al. 2011 Br J Pharmacol (PMID: 21371012) and Robinson A.P. et al. 2014 Handb Clin Neurol (PMID: 24507518).

On paragraph 4.1, the authors mention inflammatory mediators pointing out IL-1beta as an important cytokine in MS. Considering their paragraph on neutrophil-DC hybrids and GM-CSF as an important driver of their differentiation and considering the role of GM-CSF in neuroinflammation (for example, Croxford, A.L., et al, Cell (2015)), it could be interesting to add a paragraph on GM-CSF further suggesting its possible role in driving hybrid generation in MS.

On the paragraph 4.4 (Line 471), the author point out the differences between male and female and suggest the possible role for sex-differences but they do not explore. Even though it is the goal of the current review, it would be nice to see this explored since they mention it.

The review covers the global knowledge on the topic, however the recent research article published by Pavelek Z. et al. 2020 on J Clin Med (PMID: 32422897) was not cited, but considering its importance deserve to be added. Also two nice reviews published recently by Rossi B. et al. 2020 on Immunobiology (PMID: 31740077) and by Woodberry T. et al. 2018 on J Clin Med (PMID: 30513926) were nowhere mentioned. Considering that they are really well written and cover other aspects on the role of neutrophils in neurodegeneration/MS, they deserved to be cited.

Minor comments

  • Page 3, figure 1: there is a typing mistake in the word “hybrid”;
  • Line 51: The authors mention that “neutrophils are quickly activated through G-protein-coupled receptors (GPCRs)” and then they repeat the same idea with “This process of neutrophil chemotaxis is mediated by the GPCRs”.
  • Line 53: “CXC chemokine ligand-8/CXCL8” previously the authors have used () for abbreviations - CXC chemokine ligand-8 (CXCL8).
  • Line 99: there is a typing error in the word “neutrophils”;
  • Line 150: a hyphen in the. Word “post-translational” is missing;
  • Line 199: the article “the” before “T cells” is unnecessary;
  • Line 212: ... years when new cytokines and proteases … the word “ago” is probably missing;
  • Line 582: there is a change in the font style;
  • Page 16, ref 22: the publication year is not in bold.

Round 2

Reviewer 1 Report

The Authors answered properly to all my questions. I have no further comments.